# Fluorescence of Size-Fractioned Humic Substance Extracted from Sediment and Its Effect on the Sorption of Phenanthrene

**DOI:** 10.3390/ijerph16245087

**Published:** 2019-12-13

**Authors:** Mei-Sheu Shi, Wei-Shiang Huang, Liang-Fong Hsu, Yi-Lung Yeh, Ting-Chien Chen

**Affiliations:** 1Department of Civil Engineering, National Pingtung University of Science and Technology, Pingtung 91201, Taiwan; 3001@ptfire.gov.tw (M.-S.S.); yalung@mail.npust.edu.tw (Y.-L.Y.); 2Department of Environmental Science and Engineering, National Pingtung University of Science and Technology, Pingtung 91201, Taiwan; stefsun921015@gmail.com; 3Department of Applied English, Tainan University of Technology, Tainan 71002, Taiwan; td0011@mail.tut.edu.tw

**Keywords:** phenanthrene, sediment humic substances, size-fractioned HS, fluorescence quenching, sorption constants log K_HS_

## Abstract

Phenanthrene (Phe) is a toxin and is ubiquitous in the environment. The sediment humic substances (HS) that bind Phe affect the fate, transport, degradation, and ecotoxicology of Phe. This study investigated Phe sorption constants on size-fractioned HS extracted from river sediment. Fractions were identified as HHS (10 kDa to 0.45 μm), MHS (1–10 kDa), and LHS (<1 kDa). A fluorescence quenching (FQ) method was used to determine the Phe log K_HS_ on size-fractioned HS; the values ranged from 3.97 to 4.68 L/kg-C. The sorption constant (log K_HS_) is a surrogate of the binding capacity between HS and Phe, where a high log K_HS_ reduces the toxicity and degradation of Phe. The log K_HS_ values on HHS and MHS were significantly higher than the values on LHS (*p* = 0.015). The SUVA_254_ values of HHS and MHS were also significantly higher than the LHS value (*p* = 0.047), while fluorescence index (FI) and S_275–295_ values were significantly lower than the LHS values (*p* < 0.005). The HHS and MHS had a higher aromaticity and more terrestrial sources than LHS. The log K_HS_ had a significant correlation with the selected optical indicators (*p* < 0.002), which suggested that the HS-bound Phe was positively affected by high aromaticity, terrestrial sources, and HS molecular weight. The results demonstrated that optical methods successfully obtained log K_HS_ and the chemical properties of fractioned HS as well as the influenced factors of log K_HS_. Moreover, even the LHS had a capacity to bind with Phe.

## 1. Introduction

In sediment, humic substances (HS) control the sorption, fate, transport, degradation, and ecotoxicology of sediment-bound hydrophobic organic compounds (HOCs) [1,2,3,4,5,6,7]. HS is a heterogeneous organic mixture and is an important sediment component that binds HOCs. The extent of HOC binding on HS is related to the level of aromatic content, molecular weight, and hydrophobicity of HS and environmental conditions such as pH of the medium [4,5,6,7,8,9,10].

The chemical composition and structure of HS are particularly useful for studying the interaction between HOCs and HS and for identifying the underlying behavior mechanisms [5,6,7,9,10,11,12,13]. Studies have shown that HS structural characteristics are particularly complicated because of their natural complexity. The size, structure, and composition of HS vary greatly, depending on the origin and humification of the material [3,4,5,6,7].

In previous studies, HS samples were characterized by many methods, such as Nuclear Magnetic Resonance (NMR), element ratio, Gas Chromatography/Mass Spectrometry (GC/MS) [5,7,8], and ultraviolet and fluorescence spectra. Fluorescence and ultraviolet spectroscopy are very sensitive and useful techniques, which are often used to monitor river water, compost, landfill leaching water, dissolved organic matter (DOM), and HS extracted from soil and sediment [14,15,16,17,18,19,20].

Polycyclic aromatic hydrocarbon (PAH) compounds are toxic HOC substances and are ubiquitous in the environment. PAHs adsorb onto sediment, soil, and suspended particulate matter [1,2,3]. Moreover, PAHs bind with colloidal DOM isolated from river and seawater as well as humic substances extracted from soil and sediment [4,5,6,7,8,9,10,11,12,13,21,22,23]. One PAH, phenanthrene (Phe), is a moderately hydrophobic compound (the octanol–water partition coefficient, log K_OW_ = 4.57) that favors sorption in sediment HS [1,23]. However, the Phe organic carbon-normalized sorption coefficients (log K_OC_) show great variation [1,2,3,5,6,11,21,22,23].

Studies have observed that aquatic DOM/HS with different molecular weights exhibited varied aromaticity and hydrophobicity, and thus different binding affinities to contaminants [7,9,10,13,24,25,26,27,28]. The sample HS molecular weight and composition heterogeneity may provide an important contribution to binding behavior between HS and PAHs [5,6,7,9,10,11,12,13,21]. Previously, low-molecular-weight DOM/HS (<1 kDa, LMW) were assumed to have an insignificant sorption capacity on HOCs [29]. However, other studies have observed that low-molecular-weight DOM/HS had a strong sorption capacity on HOCs and heavy metals [27,29,30,31,32], even with sorption constants of LMW DOM being higher than high-molecular-weight DOM [29,30,32].

PAH compounds have a strong fluorescence intensity at a specific wavelength that was used to obtain the binding constant while PAH intensity is quenching at PAH binding to HS. Hence, a fluorescence quenching (FQ) method was used to study the sorption behavior of PAHs and HS/DOM [12,13,23,24,28,33,34,35,36,37], but little attention was given to sediment HS with different molecular weights, especially low-molecular-weight HS (<1 kDa). Furthermore, sediment HS chemical properties and sorption behavior affected by the drying processes (air-drying (AD) and freeze drying (FD)) have been less studied [22].

This study investigated the binding constants of Phe sorption on size-fractioned HS extracted from river sediment. The extracted HS solution was separated into three size fractions that were identified as HHS (10 kDa to 0.45 μm), MHS (1–10 kDa), and LHS (<1 kDa). The size-fractioned HS characteristics were examined by three optical indices, SUVA_254_, S_275–295_, and fluorescence index (FI), which were surrogates of aromaticity, molecular weight, and terrestrial sources, respectively. Phe binding constants on size-fractioned HHS, MHS, and LHS were measured with a fluorescence-quenching method. Our studies showed that the sorption constant (log K_HS_) implied a binding capacity between Phe and HS; a high log K_HS_ increased the Phe binding with HS and reduced the toxicity and degradation of Phe [29].

## 2. Research Method and Material

### 2.1. Sample Preparation

Samples of surface sediments (0–5 cm) were collected with a grab sampler from WuLo Creek (22°46′16.2″ N, 120°30′55.1″ E), Taiwan. The creek receives a large amount of wastewater discharged from livestock; hence, the sediment is rich in organic matter. One of collected sample was air-dried for one year and designated as the AD sample. Another sample, designated FD, was kept in a freezer for one year, then freeze-dried. The AD and FD samples were ground and passed through a 2.0-mm sieve. The sediment total organic carbon (TOC) content was analyzed using a Multi 3000 TOC/TN (total organic carbon/total nitrogen) analyzer (Analytik Jena, Thuringia, Germany). The sediment humic substances (HS) were extracted using 0.1 M NaOH at *w*/*v* = 1/20 ratio.

### 2.2. HS Size Fraction Separation

The extracted bulk HS (BHS) solution was adjusted to pH 7 and passed through a 0.45-μm filter. Then, the neutral BHS solutions were separated into three size fractions: 10 kDa < HHS < 0.45 μm, 1 kDa < MHS < 10 kDa, and LHS < 1 kDa, using cross-flow ultrafiltration equipment with a nominal weight molecular cutoff ceramic membrane cartridge (Filtanium, France, cut-off pore sizes 10 and 1 kDa, sequent, membrane area of 320 cm^2^), operating at a feed flow rate of 1.7–2.0 L/min. The penetration flow rates were 12 and 25 mL/min for the 10 and 1 kDa membranes, respectively. The working pressure was 5 kg/cm^2^. The BHS and three size-fractioned HS were measured with UV/Vis and fluorescence spectroscopy. Concentrations of dissolved organic carbon (DOC) for the bulk and size-fractioned HS samples were quantified by a TOC analyzer (TOC-L, Shimadzu, Tokyo, Japan).

### 2.3. UV/Vis Measurements

HS solution absorbance was measured with a UV/Vis spectrophotometer (U-2900, Hitachi, Tokyo, Japan), on a scanning wavelength of 200–800 nm. Background was corrected in accordance with the method reported by Helms et al. [17]. SUVA_254_ (L/m/mg-C) = (UV_254_/[HS]) × 100, UV_254_ (cm^−1^) was used as the UV/Vis absorbance indicator at 254 nm, and [HS] was the DOC concentration (mg-C/L) of the HS solution [15]. UV/Vis indicator S_275–295_ was the slope of absorbance at a wavelength interval of 275–295 nm [17].

### 2.4. Fluorescence Spectroscopy

A three-dimensional fluorescence excitation/emission matrix (EEM) was recorded by fluorescence spectrophotometer (F-7000, Hitachi, Tokyo, Japan) at 5 mg-C/L. Before measurement, the HS solution was adjusted to pH 7 with 0.3 N H_2_SO_4_. Fluorescent scanning conditions were as follows: excitation wavelength 200–450 nm at 5-nm increments, emission wavelength 250–550 nm at 2-nm increments, and scan rate 2400 nm/min. The spectra were obtained by subtracting an ultrapure water blank spectrum, recorded in the same conditions, to eliminate the Raman scatter peaks. The fluorescence index (FI) is the fluorescence intensity ratio of emission wavelengths at 450 and 500 nm with the excitation wavelength at 370 nm [20].

### 2.5. Fluorescence Quenching

Seven different HS concentrations were prepared (1–20 mg-C/L). Next, 1 mg/L Phe standard solution was added to each HS solution. This was followed by reciprocal shaking for 24 h (150 rpm), using 0.3 N H_2_SO_4_ solution adjusted to pH 7. The solution was measured with fluorescence spectroscopy (Hitachi, F-7000). The fluorescence intensity at Ex/Em = 250/349 nm of the HS solutions was detected. Since the UV/Vis absorbance at 254 nm was less than 0.2 cm^−1^, the inner filter effect was not corrected.

HS and the PAH sorption coefficients were calculated using the Stern–Volmer equation as Equation (1) [35]:
(1)$$\frac{F_{0}}{F}=1+K_{\mathrm{HS}}\;\left[\mathrm{HS}\right],$$
where *F* and *F*_0_ are the fluorescence intensities of the standard Phe solution with and without HS solution present, respectively. [HS] is the DOC concentration of HS (mg-C/L) and K_HS_ is the sorption coefficient (L/mg-C). It should be noted that Equation (1) assumes the Phe sorption on the HS solution had static quenching [35]. The K_HS_ was determined through the linear regression of the *F*_0_/*F* values with the DOC concentration of HS. The linear significance was dependent on whether the slope and intercept had the necessary significance (*p* < 0.05) to determine the suitability of the Stern-Volmer equation. The linear slope was the K_HS_ (L/mg–C), which was taken from logarithm to log K_HS_ (L/kg-C).

### 2.6. Statistical Analysis and Calculation of Fluorescence Data

In this study, using the R software (V 2.13.2) (R Core Team, Vienna, Austria) to calculate fluorescence and UV/Vis indicators, the R script developed by Lapworth and Kinniburgh [38] was followed. Linear regression and the difference test used the S-PLUS software (V 6.2) (Insightful Corporation, Seattle, WA, USA). The indicator difference test for the three size-fractioned HS solutions used the ANOVA test methods, and two-group data sets used the *t*-test method at significance levels of *p* < 0.05.

## 3. Results and Discussion

### 3.1. DOC Concentration and Carbon Mass Fraction of Size-Fractioned HS

The experimental sediments were both air-dried (AD) and freeze-dried (FD); the properties of the AD and FD samples are listed in Table 1. The pH values of the AD and FD samples were from 7.05 to 7.28. The AD and FD levels of organic matter (OM) and total organic carbon (TOC) were 5.55–7.90% and 1.71–3.95%, respectively. The OM and TOC concentrations were similar to the concentration in river sediment [3,27]. The OM and TOC FD concentrations were greater than the AD concentration but only slightly (*p*-values of 0.66 and 0.73, respectively). The AD-treated samples had slightly lower OM and TOC concentrations and pH, which may be attributed to the labile fraction of organic matter hydrolysis and biodegradation during the one-year drying procedure. The AD treatment produced higher hydrolysis and biodegradation than the FD treatment. Hence, AD samples generated more organic acid at a low pH value [39,40].

The DOC concentrations of BHS and the three size-fractioned HS are listed in Table 2. The BHS DOC concentrations were 297 ± 28 mg-C/L (5.94 ± 0.56 g/kg based on sediment mass) and 320 ± 48 mg-C/L (6.40 ± 0.96 g/kg based on sediment mass) for AD and FD samples, respectively; there was an insignificant difference between the two procedures (*p* = 0.51). The alkaline-extracted HS DOC concentrations in our study were higher than the water-extracted organic carbon from lake sediment reported by Xu et al. [32]. In each size-fractioned HS sample, the DOC concentration was insignificantly different between the AD and FD samples (*p*-values of 0.16–0.93).

Since the HS DOC concentrations of the two drying procedures were insignificantly different than the DOC concentrations of each size-fractioned HS, they were considered as pool samples. The order of measured DOC concentrations was 1660 ± 373 mg-C/L (HHS) > 754 ± 110 mg-C/L (MHS) > 66.8 ± 9.3 mg-C/L (LHS) (*p* < 0.001). The volume fractions for the HS separation were 1.0, 0.1, 0.09, and 0.81 for BHS, HHS, MHS, and LHS, respectively. The carbon mass balances were 95.3% ± 8.3% (AD) and 92.4% ± 15.0% (FD), which is within a reasonable range (80–120% for DOM/HS separated into different sized fractions) [27,32,41,42,43].

The carbon mass fractions were 57.1% ± 6.5%, 23.8% ± 4.4%, and 19.1% ± 4.1% for HHS, MHS, and LHS, respectively. The carbon mass of the high-molecular-weight fraction (HMW, >1 kDa) averaged 81%, which was greater than the carbon mass fraction of the water-extracted organic matter from lake sediment (52–58%) [32]. The HMW carbon mass fraction was much higher than that in river water, lake water, seawater, and estuary, which ranged from 23% to 66% in the literature [13,41,44,45]. The OM sediment continuously suffered hydrolysis and biodegradation. The low-molecular-weight labile OM fraction was readily decomposed, which left the OM high-molecular-weight carbon fraction in the sediment as reported by previous studies [39,40,46].

### 3.2. Optical Indicators

The HS and DOM optical indices provide useful information [14,15,16,17,18,19]. In this study, two UV/Vis and one fluorescence indicator were adopted to analyze the chemical properties of the size-fractioned HS: specific ultraviolet absorbance at wavelength at 254 nm (SUVA_254_) [15], spectroscopy slope between 275 and 295 nm (S_275–295_) [17], and FI [16,20].

The UV/Vis indicators of the size-fractioned HS in both AD and FD samples are listed in Table 3. In each size-fractioned HS, the indicator values were insignificantly different between the AD and FD samples (except the S_275–295_ of LHS). The AD sample S_275–295_ values were lower than the FD, which suggested that the AD sample had a higher molecular weight than the FD in the LHS samples.

In comparison, the indicator values of each of the AD and FD size-fractioned samples were calculated as a pool sample. SUVA_254_ is a surrogate for the abundance of aromatic group in HS. A large SUVA_254_ value indicates a high aromatic group content in HS [15,25]. The average SUVA_254_ values were 2.69 ± 0.69 and 2.48 ± 0.69 L/mg-C/m for HHS and MHS, respectively. They were significantly greater than the average LHS value of 1.75 ± 0.26 L/mg-C/m (*p* = 0.047). The SUVA_254_ values of the three size-fractioned HS samples were <3, which suggested that the composition of the studied HS samples contained predominantly hydrophilic substances [14].

The S_275–295_ indicator is a surrogate for the average molecular weight of HS. The indicator value was inverse-correlated with the DOM molecular weight [17,18,19]. The S_275–295_ values were 0.0120 ± 0.0004 and 0.0125 ± 0.0014 for HHS and MHS, respectively. They were significantly lower than the S_275–295_ value of LHS—0.0165 ± 0.0014 (*p* = 0.005). The SUVA_254_ and S_275–295_ indicator values had a strong negative correlation (*r* = −0.73, *p* < 0.001).

The FI is an indicator used to discriminate the DOM sources (e.g., microbial or terrestrial) [16,20]. The FI values were 1.47 ± 0.03 and 1.47 ± 0.02 for HHS and MHS, respectively. They were significantly lower than the FI value of LH—1.78 ± 0.04 (*p* = 0.005). The indicator results suggested that HHS and MHS had more contribution from terrestrial sources than LHS. The UV/Vis and fluorescence indicators showed that the three size-fractioned HS samples primarily contained hydrophilic substances of aquatic media and microbial origin, as well as a low degree of humification [14,16,18,19,20].

The HS indicators were comparable to water- and alkaline-extracted organic matter from river sediment, such as SUVA_254_ values that ranged from 0.2 to 3.7 L/mg-C/m, as reported by previous studies [5,6,7,27], The reported FI values ranged from 1.3 to 2.3, as reported by previous studies [6,7,16,27,32]. The S_275__–295_ values were higher than the alkaline-extracted organic matter, which ranged from 0.0074 to 0.0095, as reported by Hur et al. [5]. The S_275__–295_ values were lower than the water-extracted organic matter, which ranged from 0.017 to 0.018, as reported by Xu et al. [32].

There are few reported optical indicators for size-fractioned HS. Xu et al. [32] discussed the optical indicators of high-molecular-weight organic matter (1 kDa to 0.45 μm, HMW) and low-molecular-weight organic matter (<1 kDa, LMW) extracted from river sediment. The HMW SUVA_254_ values ranged from 2.05 ± 0.03 to 5.12 ± 0.07 L/mg-C/m and the those of the LMW ranged from 8.25 ± 0.14 to 14.60 ± 0.25 L/mg-C/m. The SUVA_254_ values were higher than our study and the HMW values were higher than LMW, which is the reverse of our study.

In the Xu et al. [32] study, the HMW S_275__–295_ values were 0.009 ± 0.001 and 0.0118 ± 0.001 and the LMW values were 0.028 ± 0.002 and 0.0188 ± 0.001. The S_275__–295_ values were comparable to our results and HMW had low S_275__–295_ values. The HMW FI values were 1.25 ± 0.02 and 1.18 ± 0.02 and the LMW FI values were 1.52 ± 0.03 and 1.34 ± 0.02. The FI values were lower than in our study but had a similar trend; the HMW had lower FI values than the LMW. In our study, the HMW HS had more aromaticity and terrestrial sources than the LMW. However, the water-extracted organic matter as reported in Xu et al. [32] showed that the HMW had low aromaticity and more terrestrial sources. The HS sources, environmental condition, and HS extraction method affected the HS indicator values.

### 3.3. Sorption Constants between HS and Phe

The Phe sorption constants on the size-fractioned HS were measured with the FQ method. The FQ method has been previously used to test sorption constants between several PAHs onto HS [12,13,23,24,28,33,34,35,36,37]. Figure 1a,b shows the linear regression of fluorescence intensity *F*_0_/*F* ratios of Phe with HHS, MHS, and LHS concentrations fitted with the Stern–Volmer equation for both AD and FD samples. The *F*_0_/*F* ratios of Phe increased linearly and followed the increasing HS concentrations, which indicated the intensity (F) decreased and followed the extent of interaction between Phe and HS. This was attributed to the sorption between Phe and HS, which decreased the fluorescence intensity of Phe.

Three size fractions and two drying methods were conducted. All tests were run in triplicate for a total of 18 tests. The slopes of the linear regression were the sorption constant K_HS_ values (L/mg-C), which were transferred to log K_HS_ (L/kg-C). In total, 16 out of 18 FQ tests had a significant linear relationship. In each HS size fraction, the validated FQ tests were 6, 5, and 5 samples for HHS, MHS, and LHS, respectively. The average and one standard deviation of log K_HS_ for each size-fractioned HS are listed in Table 4. The average intercepts of regression ranged from 0.79 to 0.95 and *r*^2^ ranged from 0.62 to 0.90 for the three size-fractioned HS. The log K_HS_ values of Phe sorption on the three size-fractioned HS were 3.97–4.68 L/kg-C. Several studies tested HS extracted from soils and sediments. The organic carbon normalized sorption constants log K_HS_ of Phe sorption on HS ranged from 3.76 to 6.13 [1,2,3,5,6,21,22,34,36,37]. These results are comparable to the sorption constants in this study.

In each size-fractioned HS, the log K_HS_ values were not significantly different between the AD and FD samples (*p* = 0.17–0.87). However, Phe had higher log K_HS_ sorption values onto HHS (4.43 ± 0.21) and MHS (4.41 ± 0.21) than sorption onto LHS (4.04 ± 0.09) (*p* = 0.015). Several studies [7,12,13,22,24] have reported that the HOCs had a higher sorption capacity onto higher molecular weight DOM because it had a higher aromatic content and higher extent of humification. Although, LHS had a lower log K_HS_ than HHS and MHS. It was noteworthy that generally, LMW (<1 kDa) HS was considered without sorption capacity on HOCs. A few studies have observed that LMW HS/DOM had sorption on HOCs and heavy metal capacity [28,29,30,31,32,33] even higher than HMW [29,30,32].

### 3.4. Correlation of log K_HS_ with Indicators

The total log K_HS_ value was 4.30 ± 0.23 and the log K_OW_ of Phe was 4.57 [1,23]. The value of log K_HS_ was close to octanol–water partition coefficient (log K_OW_) values, which suggested that hydrophobicity was an important factor that impacted the sorption behavior of Phe. The HOC log K_OW_ and log K_OC_ have been observed to have a positive relationship as reported by previous studies [47,48,49,50]. However, the independent relationship also has been observed in several studies of HOC sorption onto HS [51,52]. This suggests that the simple partitioning of PAH hydrophobicity would not accurately describe the sorption of HOCs to HS, where the chemical composition and structure as well as the HS molecular weight can be the important factors that influence the sorption behavior (as reported by previous studies [5,52]).

The correlation between log K_HS_ and the indicators may reflect the important factors that impact the extent of interaction between HS and Phe. Figure 2a–c shows the linear relationships of the Phe log K_HS_ with optical indices (SUVA_254_, S_275–295_ and FI). SUVA_254_ had an intermediate positive correlation with the Phe log K_HS_ (*r*^2^ = 0.50, *p* = 0.002) (Figure 2a). Both S_275–295_ and FI values had a significantly negative correlation with log K_HS_ (*r*^2^ = 0.76, *r*^2^ = 0.67, for S_275–295_ and FI, respectively, *p* < 0.001). Low S_275–295_ value and FI values indicated high molecular weight and greater contribution of terrestrial sources in HS (Figure 2b,c).

More aromatic functional groups, terrestrial sources, and high molecular weight in HS had a positive effect on Phe sorption onto HS. Previous studies have shown that aromaticity, terrestrial sources, and molecular weight had a positive correlation with sorption constants of HOCs onto HS and DOM [5,6,7,10,11,13,34]. Generally, terrestrial sources had more aromatics and high-molecular-weight HS had more intramolecular charge transfer capability, which increased the sorption capacity of HS-Phe [5,7,10,11].

## 4. Conclusions

DOC concentrations and optical indicators, for both total and sized HS, were insignificantly different for AD or FD samples. Furthermore, the binding constants were insignificantly different between the AD and FD samples. However, HHS and MHS had higher binding constants, log K_HS_, than LHS; the HHS and MHS had higher aromaticity, terrestrial sources, and molecular weight than LHS. The high binding capacity implied that HS-bound Phe had low toxicity to microorganisms but a long resistance in the environment. The results demonstrated that optical methods successfully obtained log K_HS_ and chemical properties of fractioned HS as well as the influenced factors of log K_HS_. Moreover, even the LHS (<1 kDa) had significant binding capacity with Phe.

## Figures and Tables

**Figure 1 ijerph-16-05087-f001:**
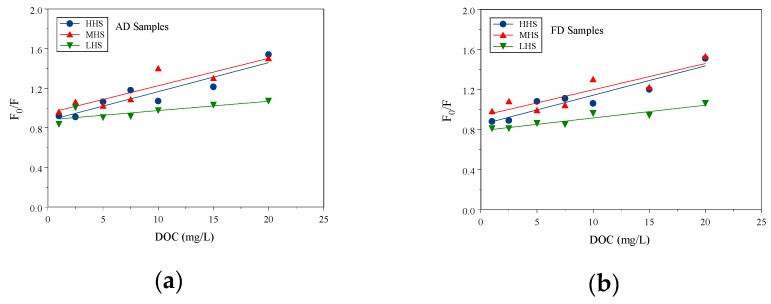
*F*_0_/*F* ratios with DOC concentration and linear regression curves for (**a**) AD sample and (**b**) FD sample.

**Figure 2 ijerph-16-05087-f002:**
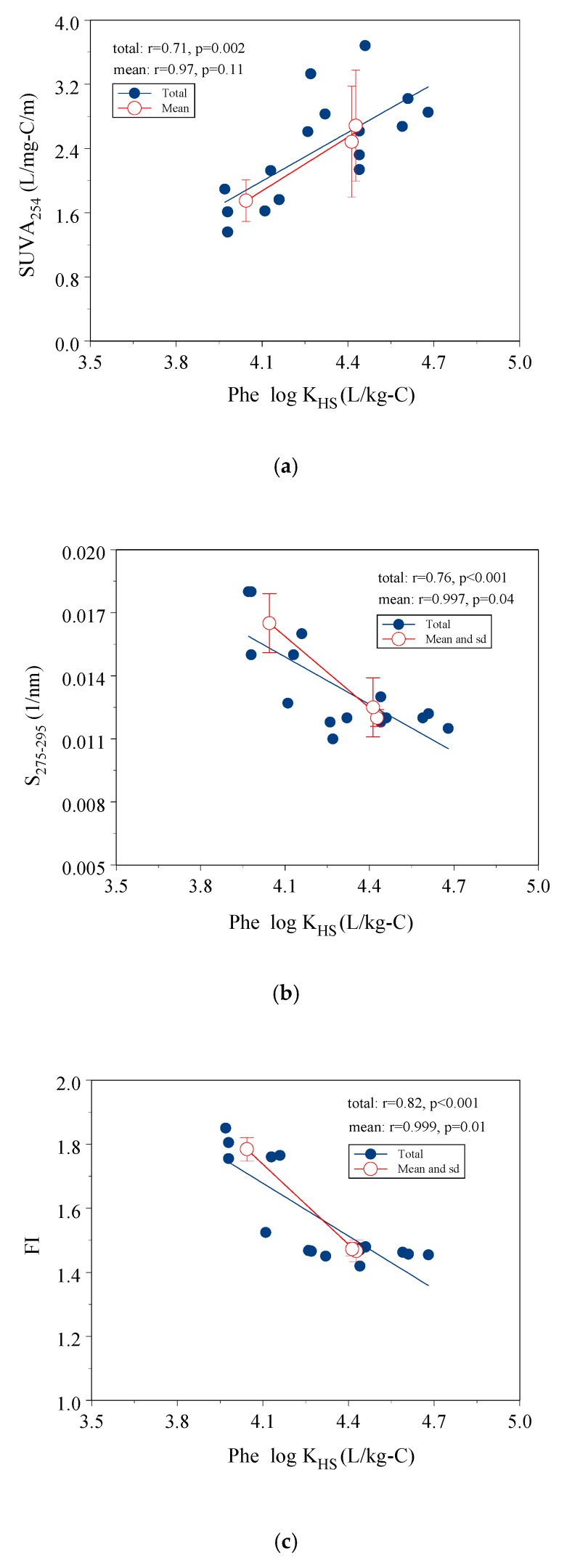
Correlation of Phe log K_HS_ with three optical indicators, (**a**) SUVA_254_; (**b**) S_275–295_; (**c**) FI.

**Table 1 ijerph-16-05087-t001:** Sediment chemical properties for both air-dried (AD) and freeze-dried (FD) samples.

Samples	pH	OM (%)	TOC (%)
AD	7.14 ± 0.12	6.60 ± 1.03	2.51 ± 0.69
FD	7.22 ± 0.03	7.06 ± 1.36	2.71 ± 0.70

Organic matter (OM); total organic carbon (TOC).

**Table 2 ijerph-16-05087-t002:** Measured dissolved organic carbon (DOC) concentrations of bulk humic substances (HS) and three size-fractioned HS for both AD and FD samples.

Samples	BHS mg/L	HHS mg/L	MHS mg/L	LHS mg/L
AD	276–329	1451–1833	638–752	59–78
FD	279–370	1108–2220	728–946	52–71

Molecular weight of humic substances: BHS (<0.45 μm), HHS (10 kDa—0.45 μm), MHS (1 kDa—10 kDa), and LHS (<1 kDa).

**Table 3 ijerph-16-05087-t003:** Optical indicators of three size-fractioned HS for both AD and FD samples.

Samples (MW)	SUVA_254_ (L/mg-C/m)	S_275–295_	FI
AD_HHS (10 kDa to 0.45 μm)	2.26 ± 0.62 ^a^	0.0121 ± 0.0006 ^a^	1.47 ± 0.05 ^a^
FD_HHS (10 kDa to 0.45 μm)	3.11 ± 0.54 ^a^	0.0120 ± 0.0002 ^a^	1.47 ± 0.01 ^a^
AD_MHS (1–10 kDa)	2.95 ± 0.34 ^a^	0.0117 ± 0.0006 ^a^	1.46 ± 0.01 ^a^
FD_MHS (1–10 kDa)	2.02 ± 0.66 ^a^	0.0133 ± 0.0016 ^a^	1.49 ± 0.02 ^a^
AD_LHS (<1 kDa)	1.75 ± 0.38 ^b^	0.0153 ± 0.0006 *^,b^	1.76 ± 0.01 ^b^
FD_LHS (<1 kDa)	1.75 ± 0.14 ^b^	0.0177 ± 0.0006 ^b^	1.81 ± 0.04 ^b^

MW: molecular weight; * indicates AD and FD were significantly different (*p* < 0.05); ^a,b^ indicate that the sized fractions had significant difference (*p* < 0.005).

**Table 4 ijerph-16-05087-t004:** The sorption constants (log K_HS_) of phenanthrene (Phe) with three size-fractioned HS for both AD and FD samples.

Samples	HHS	MHS	LHS
AD	4.41 ± 0.29 (3) *	4.39 ± 0.17 (3)	4.09 ± 0.10 (3)
FD	4.44 ± 0.17 (3)	4.44 ± 0.00 (2)	3.97 ± 0.01 (2)

* The sample number is in parentheses.

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
