# Peer review of "Fluorescence of Size-Fractioned Humic Substance Extracted from Sediment and Its Effect on the Sorption of Phenanthrene"

_ijerph, 2019, doi:10.3390/ijerph16245087_

Round 1

Reviewer 1 Report

Ijerph-653253-comments

November 21, 2019

General comments:

The methods and data in this paper are very convincing, however the authors aren’t giving me a reason to read it.

The abstract needs to get me interested enough to read the whole article. It should be shorter but with more explanation of why your study is important. You should state that Phe is a toxin and that humic substances will impact the fate and transport. You want to know if the binding constants between HS and Phe are different for different molecular weight fractionations, and you want to show that you can use optical methods to detect these differences. Then you need to conclude with a statement of why your findings are important.

Please see the line-by-line comments for some suggestions.

The Introduction section didn’t tell me how the HS impact fate and transport, i.e. do they make it more bio-available or last longer in the water? Also, I think this would be the place to state what you are measuring. Did you measure the fluorescence quenching of humics or of Phe? I didn’t see that until nearly the end of the paper.

The Research Methods and Material section was straight forward.

The Results and Discussion section mixed your results with literature results in a way that was unclear. Sometimes I was left wondering if the data was from the reference or your work. I think you can do some simple edits to make that distinction. In addition, given the variability in composition of HS from different sites, why would you expect to be able to correlate your results to other studies? You should identify the source of humics in the other studies that you reference.

The Conclusion was really just a summary. I would have like to have seen a statement about how your results are important. In addition, that statement should also be put into the abstract.

Line-by-line

14 – You can leave out “that exists”.

15 – In what way does HS binding affect Phe fate and transport?

18 – I suggest “determine” instead of “investigate”.

19 – To shorten the abstract you could leave out the range of values since your giving values in the next sentence.

21 – You didn’t give the written out version of “SUVA254”

41 – define Kow

111 – I suggest, “correction was not needed.”

119 – It might help readability if you don’t use abbreviations unless they are going to be used throughout the article. In this case, the S-V abbreviation is just something that the reader has to figure out.  This article has a large number of abbreviations and code letters.  Be kind to the reader, please.

179 – You didn’t define “S275-295”.

189  - Did you mean “aquatic media”?

191 – Did you mean “comparable” instead of “compatable”? Comparable seems better to me.

201 – The source of comparison data is crucial considering the variability of humic substances in water.

206 – I don’t know what “WEOM” is.

209 – As a reader, it would be nice to see another column with the MW fraction and drying summarized for me right in this table. Then, at a glance, I would see what’s going on.

218 – This is the first time in the article that I realized you were looking at the quenching of the Phe fluorescence. Since humic substances have fluorescence signatures, I thought that is what you were measuring.

219 – This sentence is awkward. You could just reverse it to say “three size fractions and six concentrations for a total of 18 tests.” Then a separate sentence, “All tests were conducted in triplicate.”

234 – This sentence is an example of how you mix your data with the literature data. It would help me, as the reader, to be clearer about this. Separate sentences might be all you need.

253 – I don’t know why you are using the individual values from the Khs rather than the means from before. It makes me wonder what the graphs look like if the mean and range were plotted.

269 – Sentence needs to be clarified. Maybe just flip it around, i.e. “DOC, optical indicators, and size fractionation were not significantly different for AD or FD.

272 – However, I’d rather have you lead with what was significant in the conclusion and then move on to why this is important. After reading your article several times, I still don’t know why we need to know about absorption coefficients for Phe and HS. I bet you could look at the abstracts for some of the articles in your references to see that, for example the articles by McKnight et al in L&O of Sun et al in Chemosphere.

Author Response

Review #1

General comments:

#1-1 The methods and data in this paper are very convincing, however the authors aren’t giving me a reason to read it.

Response #1-1:

Thank you for the comments and suggestions. The general and specific comments and suggesstions for this manuscript have been improved and modified, point by point. Please see the line-by-line comments for some suggestions.

#1-2 The abstract needs to get me interested enough to read the whole article. It should be shorter but with more explanation of why your study is important. You should state that Phe is a toxin and that humic substances will impact the fate and transport. You want to know if the binding constants between HS and Phe are different for different molecular weight fractionations, and you want to show that you can use optical methods to detect these differences. Then you need to conclude with a statement of why your findings are important.

Response #1-2: Thank you for the comments and suggestions. The abstract has been modified as the reviewer has suggested.

Phenanthrene (Phe) is a toxin and ubiquitous in the environment. The sediment humic substances (HS) that bind Phe affect the fate, transport, degradation, and ecotoxicology of Phe. This study investigated Phe sorption constants on size-fractioned HS extracted from river sediment. Fractions were identified as HHS (10 kDa – 0.45 μm), MHS (1 -10 kDa), and LHS (< 1 kDa). A Fluorescence Quenching (FQ) method was used to determine the Phe log KHS on size-fractioned HS; the values ranged from 3.97 to 4.68 L/kg-C. The sorption constant (log KHS) is a surrogate of the binding capacity between HS and Phe where a high log KHS reduces the toxicity and degradation of Phe. The log KHS values on HHS and MHS were significantly higher than the values on LHS (p=0.015). The SUVA254 values of HHS and MHS were also significantly higher than the LHS value (p=0.047), while fluorescence index (FI) and S275-295 values were significantly lower than the LHS values (p<0.005). The HHS and MHS had a higher aromaticity and more terrestrial sources than LHS. The log KHS had a significant correlation with the selected optical indicators (p<0.002), which suggested that the HS-bound Phe was positively affected by high aromaticity, terrestrial sources and HS molecular weight. The results demonstrated that optical methods successfully obtained log KHS and the chemical properties of fractioned HS as well as the influenced factors of log KHS. Moreover, even the LHS had a capacity to bind with Phe.

#1-3 The Introduction section didn’t tell me how the HS impact fate and transport, i.e. do they make it more bio-available or last longer in the water? Also, I think this would be the place to state what you are measuring. Did you measure the fluorescence quenching of humics or of Phe? I didn’t see that until nearly the end of the paper.

Response #1-3: Thank you for the comments and suggestions. The Introduction Section has had a statement added on how the sorption constant influences toxicity and degradation at Phe bound to HS. The modified Introduction is as follows.

Introduction

In sediment, humic substances (HS) control the sorption, fate, transport, degradation, and ecotoxicology of sediment-bound hydrophobic organic compounds (HOCs) [1-7]. HS is a heterogeneous organic mixture and is an important sediment component that binds HOCs. The extent of HOC binding on HS is related to the level of aromatic content, molecular weight, and hydrophobicity of HS, and environmental conditions such as pH of the medium [4-7,10,14,16].

The chemical composition and structure of HS is particularly useful for studying the interaction between HOCs and HS and for identifying the underlying behavior mechanisms [5-7,12-16]. Studies have shown that HS structural characteristics are particularly complicated because of their natural complexity. The size, structure, and composition of HS vary greatly, depending on the origin and humification of the material [3-7].

In previous studies, HS samples were characterized by many methods such as NMR, element ratio, GC/MS [5,7,10], and ultraviolet and fluorescence spectra. Fluorescence and ultraviolet spectroscopy are very sensitive and useful techniques, which are often used to monitor river water, compost, landfill leaching water DOM and HS extracted from soil and sediment [18-24].

Polycyclic aromatic hydrocarbon (PAH) compounds are toxic HOC substances and are ubiquitous in the environment. PAHs adsorb onto sediment, soil, and suspended particulate matter [1-3]. Moreover, PAHs bind with colloidal dissolved organic matter (DOM) isolated from river and seawater as well as humic substances extracted from soil and sediment [4-7,9-17]. One PAH, phenanthrene (Phe), is a moderately hydrophobic compound (the octanol/water partition coefficient, log KOW = 4.57) that favors sorption in sediment HS [1,17]. However, the Phe organic carbon-normalized sorption coefficients (log KOC) show great variation [1-3,5,6,9,11,12,17].

Studies have observed that aquatic DOM/HS with different molecular weight exhibited varied aromaticity and hydrophobicity, and thus different binding affinities to contaminants [7,14-16,25-29]. The sample HS molecular weight and composition heterogeneity may provide an important contribution to binding behavior between HS and PAHs [5-7,9,12-16]. Previously, low molecular weight DOM/HS (< 1 kDa, LMW) was assumed to have an insignificant sorption capacity on HOCs [8]. However, other studies have observed that LMW DOM/HS had a strong sorption capacity on HOCs and heavy metals [8,28,30-32], even with sorption constants of LMW DOM higher than HMW DOM [8,30,32].

PAH compounds have a strong fluorescence intensity at a specific wavelength that was used to obtain the binding constant while PAH intensity is quenching at PAH binding to HS. Hence, a fluorescence quenching (FQ) method was used to study the sorption behavior of PAHs and HS/DOM [13,15,17,25,29,33-37], but little attention was given to sediment HS with different molecular weights, especially low molecular weight HS (<1 kDa). Furthermore, sediment HS chemical properties and sorption behavior affected by the drying processes (air-drying (AD) and freeze drying (FD)) have been less studied [11].

This study investigated the binding constants of Phe sorption on size-fractioned HS extracted from river sediment. The extracted HS solution was separated into three size fractions that were identified as HHS (10 kDa – 0.45 μm), MHS (1 -10 kDa), and LHS (< 1 kDa). The size-fractioned HS characteristics were examined by three optical indices, SUVA254, S275-295, and fluorescence index (FI), which were surrogates of aromaticity, molecular weight, and terrestrial sources. Phe binding constants on size-fractioned HHS, MHS, and LHS were measured with a fluorescence-quenching method. Our studies showed that the sorption constant (log KHS) implied a binding capacity between Phe and HS; a high log KHS increased the Phe binding with HS and reduced the toxicity and degradation of Phe [8].

#1-4 The Research Methods and Material section was straight forward.

Response #1-4: Thank you for the comments.

#1-5 The Results and Discussion section mixed your results with literature results in a way that was unclear. Sometimes I was left wondering if the data was from the reference or your work. I think you can do some simple edits to make that distinction. In addition, given the variability in composition of HS from different sites, why would you expect to be able to correlate your results to other studies? You should identify the source of humics in the other studies that you reference.

Response #1-5: Thank you for the comments and suggestions.

We have performed several edits to distinguish the results of our study and the literature. The changes to the improved paragraphs have been changed to a blue color in the revised manuscript.

The chemical composition and structure of HS was important for the interaction of HOC and HS as was shown by many studies appearing in the cited references. In our study, the optical indicators were used as surrogates of HS chemical composition and structure. We used the correlation between HS chemical composition and structure (optical indicators) and log KHS to obtain the control factors of Log KHS. The correlations imply that the results in this study can be applied to the HS sources from other sites.

#1-6 The Conclusion was really just a summary. I would have like to have seen a statement about how your results are important. In addition, that statement should also be put into the abstract.

Response #1-6: Thank you for the comments and suggestions.

The conclusion has been intensively revised. We have added the importance of this study, and this statement has been added to the Abstract. The revised Conclusion is as follows.

“DOC concentrations and optical indicators, for both total and sized HS were insignificantly different for AD or FD samples. Furthermore, the binding constants were insignificantly different between the AD and FD samples. However, HHS and MHS had higher binding constants, log KHS, than LHS; the HHS and MHS had higher aromaticity, terrestrial sources and molecular weight than LHS. The high binding capacity implied that HS bound Phe had low toxicity to microorganisms, but a long resistance in the environment. The results demonstrated optical methods successfully obtained log KHS and chemical properties of fractioned HS as well as the influenced factors of log KHS. Moreover, even the LHS (<1 kDa) had significant binding capacity with Phe.”

Line-by-line

#1-7 14 – You can leave out “that exists”.

Response #1-7: Thank you for the comments and suggestions.

Line 14 the “that exists” has been deleted.

#1-8 15 – In what way does HS binding affect Phe fate and transport?

Response #1-8: Thank you for the comment.

The binding constant (log KHS) is a surrogate of the binding capacity between dissolved HS and Phe; the high log KHS increased the Phe binding with HS and reduced the toxicity and degradation of Phe [8].

#1-9 18 – I suggest “determine” instead of “investigate”.

Response #1-9: Thank you for the comments and suggestions.

Line 19 Theinvestigate” has changed to “determine”.

#1-10 19 – To shorten the abstract you could leave out the range of values since your giving values in the next sentence.

Response #1-10: Thank you for the comments and suggestions.

Lines 20-24: The Phe-HS log KHS values ranging from 3.97 to 4.68 L/kg-C have been deleted. In addition, in order to shorten the abstract, the average values of log KHS, and indicators have been ignored in the revised manuscript. However, the statement how the log KHS affects the Phe toxicity and degradation have been added. In addition, a statement about the importance of the results has been added to the Abstract.

#1-11 21 – You didn’t give the written out version of “SUVA254

Response #1-11: Thank you for the comment.

Since we are limiting the words in the Abstract, the statement how the log KHS affected the Phe toxicity and degradation has been added. A statement about the importance of the results has been added to the Abstract. The difference of indicators in sized HS was reported.

#1-12 41 – define Kow

Response #1-12: Thank you for the comment.

Line 45 adds “(the octanol/water partition coefficient, log KOW = 4.57)”.

#1-13 111 – I suggest, “correction was not needed.”

Response #1-13:Line 116 “correction” has been deleted.

#1-14 119 – It might help readability if you don’t use abbreviations unless they are going to be used throughout the article. In this case, the S-V abbreviation is just something that the reader has to figure out. This article has a large number of abbreviations and code letters. Be kind to the reader, please.

Response #1-14: Thank you for the comments and suggestions.

In the revised manuscript we have changed the abbreviations unless they are used throughout the article.

Line 125 the “the S-V equation” has been changed to “Stern-Volmer equation”

#1-15 179 – You didn’t define “S275-295”.

Response #1-15: Thank you for the comment.

Line 185-186 defines S275-295.The S275-295 indicator is a surrogate for the average molecular weight of HS. The indicator value was inverse- correlated with the DOM molecular weight [21-23].”

In addition the calculation of S275-295 is listed in Line 101 “UV/Vis indicator S275-295 was the slope of absorbance at a wavelength interval of 275-295 nm [21].”

#1-16 189 - Did you mean “aquatic media”?

Response #1-16: Thank you for the comment.

Line 195 “median aquatic” has been changed to “aquatic media”

#1-17 191 – Did you mean “comparable” instead of “compatable”? Comparable seems better to me.

Response #1-17: Thank you for the comment.

Line 197 “compatible” has been changed to “comparable”

#1-18 201 – The source of comparison data is crucial considering the variability of humic substances in water.

Response #1-18: Thank you for the comment.

UV/Vis slope indicator S275-295 is a surrogate of DOM molecular weight. The UV/Vis spectrum depends on the composition of HS/DOM. Generally, the different molecular weights have different S275-295 values. In this study, the HHS (10 kDa-0.45 μm) and MHS (1-10 kDa) had similar S275-295 values (0.0120±0.0004 and 0.0125±0.0014, respectively). They were significantly lower than the values of LHS (0.0165±0.0014). The indicator values of SUVA254 and FI also had a similar tendency. Moreover, the log KHS of HHS and MHS had similar values, which were significantly higher than the LHS values. These results indicated that HHS and MHS had a similar composition. Although, the HHS and MHS had different molecular weights, they had similar S275-295 values, which signified that the composition of HHS and MHS were similar. The significant correlation between HS and S275-295 suggested S275-295 was a good indicator to obtain the composition characteristic.

Original Line 193. We agree that the reported S275-295 values of water-extracted organic matter, which ranged from 0.0074 to 0.029, were too wide for the discussion of similarity. We have changed the comparison S275-295 to literature data as follows.

Lines 199-203. “The S275-295 values were higher than the alkaline-extracted organic matter, which ranged from 0.0074 to 0.0095, as reported by Hur et al. [5]. The S275-295 values were lower than the water-extracted organic matter, which ranged from 0.017 to 0.018, as reported by Xu et al. [32]. In addition, the S275-295 values of LMW DOM ranged from 0.019 to 0.028. They were significantly higher than the HMW DOM, which ranged from 0.009 to 0.012, as reported by Xu et al. [32].”

#1-19 206 – I don’t know what “WEOM” is.

Response #1-19: Thank you for the comment.

Line 215 “WEOM” has been changed to “water-extracted organic matter. It has also been changed in Lines 157 and 195.

#1-20 209 – As a reader, it would be nice to see another column with the MW fraction and drying summarized for me right in this table. Then, at a glance, I would see what’s going on.

Response #1-20: Thank you for the comments and suggestions.

Table 3 has been changed to show the range of MW, the difference of the indicator among the size-fractioned HS, and the difference between AD and FD samples, with a superscript. The modified Table 3 is as follows:

Table 3. Optical indicators of three size-fractioned HS for both AD and FD samples.

Samples (MW)

SUVA254 (L/mg-C/m)

S275-295

FI

AD_HHS (10 kDa-0.45 μm)

2.26±0.62a

0.0121±0.0006 a

1.47±0.05 a

FD_HHS (10 kDa-0.45 μm)

3.11±0.54 a

0.0120±0.0002 a

1.47±0.01 a

AD_MHS (1-10 kDa)

2.95±0.34 a

0.0117±0.0006 a

1.46±0.01 a

FD_MHS (1-10 kDa)

2.02±0.66 a

0.0133±0.0016 a

1.49±0.02 a

AD_LHS (<1 kDa)

1.75±0.38 b

0.0153±0.0006* b

1.76±0.01 b

FD_LHS (<1 kDa)

1.75±0.14 b

0.0177±0.0006 b

1.81±0.04 b

MW: molecular weight; * indicates AD and FD were significantly different (p< 0.05). a,b indicated the sized fractions had significant difference (p<0.005).

#1-21 218 – This is the first time in the article that I realized you were looking at the quenching of the Phe fluorescence. Since humic substances have fluorescence signatures, I thought that is what you were measuring.

Response #1-21: Thank you for the comments and suggestions.

Line 62-63 the principal of fluorescence quenching has been added to the Introduction Section. “PAH compounds had a strong fluorescence intensity at a specific wavelength, which was used to obtain the binding constant while PAH intensity is quenching at PAH binding to HS.

#1-22 219 – This sentence is awkward. You could just reverse it to say “three size fractions and six concentrations for a total of 18 tests.” Then a separate sentence, “All tests were conducted in triplicate.”

Response #1-22: Thank you for the comments and suggestions.

Lines 230-232 has been changed to “Three size fractions and two drying methods were conducted. All tests were run in triplicate for a total of 18 tests”.

#1-23 234 – This sentence is an example of how you mix your data with the literature data. It would help me, as the reader, to be clearer about this. Separate sentences might be all you need.

Response #1-23: Thank you for the comments and suggestions.

Lines 244-247 was the results of literature data. The sentence has been changed to: “Several studies [7,11,13,15,25] have reported that the HOCs had a higher sorption capacity onto higher molecular weight DOM because it had a higher aromatic content and higher extent of humification. In addition, the whole revised manuscript has been edited to distinguish the results from references and our results. The changed paragraphs are highlighted with a blue color.

#1-24 253 – I don’t know why you are using the individual values from the Khs rather than the means from before. It makes me wonder what the graphs look like if the mean and range were plotted.

Response #1-24: Thank you for the comments and suggestions.

The mean and standard deviation of log KHS for sized HS have been added to Figs. 2a-2c. The correlation between log KHS and the indicators used the individual values from the log KHS because the log KHS and indicators were varied and uncertain. Moreover, this study only contained three size-fractioned HS. The sample numbers were small; hence, the correlation between log KHS and the indicators used the individual values from the log KHS.

Figure 2. Correlation of Phe log KHS with three optical indicators, (a) SUVA254 (b) S275-295 (c) FI.

#1-25 269 – Sentence needs to be clarified. Maybe just flip it around, i.e. “DOC, optical indicators, and size fractionation were not significantly different for AD or FD.

Response #1-25: Thank you for the comments and suggestions.

Line 284-285 The sentence has been changed to ”DOC concentrations and optical indicators, for both total and sized HS, were insignificantly different between the AD and FD samples.”

#1-26 272 – However, I’d rather have you lead with what was significant in the conclusion and then move on to why this is important. After reading your article several times, I still don’t know why we need to know about absorption coefficients for Phe and HS. I bet you could look at the abstracts for some of the articles in your references to see that, for example the articles by McKnight et al in L&O of Sun et al in Chemosphere.

Response #1-26: Thank you for the comments and suggestions.

The meaning and importance of log KHS has been addressed in the Abstract and Introduction. The modified Conclusion is as follows.

“DOC concentrations and optical indicators, for both total and sized HS, were insignificantly different between the AD and FD samples. Furthermore, the binding constants were insignificantly different between the AD and FD samples. However, HHS and MHS had higher binding constants, log KHS, than LHS; the HHS and MHS had a higher aromaticity, terrestrial sources and molecular weight than LHS. The high binding capacity implied that HS bound Phe had low toxicity to microorganisms, but a long resistance in the environment. The results demonstrated optical methods successfully obtained log KHS and chemical properties of fractioned HS as well as the influenced factors of log KHS. Moreover, even the LHS had significant binding capacity with Phe.”

Reviewer 2 Report

The fluorescence study of size-fractioned humic substances was very interesting. However, the explanation of the detail of experiment and discussion can be improved.

Comments

1.  In the sample preparation section. Did you conduct desalination of HS in the process of purification? In my opinion, desalination is necessary for the accurate discussion of the interaction of humic substances. If not, the unneeded reason should be described. 

2. At least, elemental composition of HHS, MHS, and LHS is necessary as the characterization of humic substances and its impurities.

3  In the section of HS size fraction separation.  The value of pH in the separation should be shown. You know, humic substances can be divide 3 categories, fulvic acid, humic acid, and humin. Humic acid must be precipitated in acidic condition, therefore, the fraction separetaion result may be affected by solution pH.

4. About the S275-295 analysis.  In line 179, you said this value correlated with DOM molecular weight.  Then, you said the value of HHS and MHS was lower than LHS. From my point, it is a matter of course since you separate them by filters. So my question is, what is the significance of this mesurement? You said "the S275-295 values were similar to water-extracted organic matter, which ranged from 0.0074 to 0.029" in line 193, but that range is too wide for the discussion of similarity in my opinion. 

Author Response

Review #2

#2-1 The fluorescence study of size-fractioned humic substances was very interesting. However, the explanation of the detail of experiment and discussion can be improved.

Response #2-1: Thank you for the comments and suggestions.

#2-2 In the sample preparation section. Did you conduct desalination of HS in the process of purification? In my opinion, desalination is necessary for the accurate discussion of the interaction of humic substances. If not, the unneeded reason should be described.

Response #2-2: Thank you for the comments and suggestions.

In this study, the FQ method was used to obtain the binding constants of the size-fractioned HS and Phe. The HS chemical properties were important factors that impacted sorption capacity; hence, we focused on the optical indicators and how they affected the binding capacity. We agree that desalination of HS is an important factor for some study areas; for example, the fouling in membrane filtration. However, the desalination of HS has not been studied in the binding of PAH to HS, for example in the cited references [13,15,17,25,29,33-37].

#2-3 At least, elemental composition of HHS, MHS, and LHS is necessary as the characterization of humic substances and its impurities.

Response #2-3: Thank you for the comments and suggestions.

We agree that the elemental composition is important for characterization of humic substances and its impurities. Several previous studies have analyzed elemental composition of whole solid particulates on sorption. Generally, the sorption constant log Kd has a positive correlation to total organic carbon content; for example, the cited references [1-3,9]. In addition, a few studies have analyzed elemental composition of dissolved humic substances extracted from sediment and soil. However, only a few studies have shown that the elemental compositions of humic substances were one of the control factors that affected the binding constants between PAH and HS, such as studied by Hur et al. [5]. In our study, we focused on the difference between the binding coefficients of sized HS and Phe, by drying methods and sized HS with an optical method. Hence, we did not analyze the elemental composition characteristics of sized HS.

#2-4 In the section of HS size fraction separation. The value of pH in the separation should be shown. You know, humic substances can be divide 3 categories, fulvic acid, humic acid, and humin. Humic acid must be precipitated in acidic condition, therefore, the fraction separation result may be affected by solution pH.

Response #2-4: Thank you for the comments and suggestions.

Line 79 described the bulk HS extraction process. “The sediment humic substances (HS) were extracted using 0.1 M NaOH at w/v=1/20 ratio.” The extracted bulk HS solution was at alkaline condition. Before the bulk HS was separated into 3 size-fractioned HS, the bulk was adjusted to pH 7.

Line 81 has been modified to “The extracted bulk HS (BHS) solution was adjusted to pH 7 and passed through a 0.45 μm filter. Then the neutral BHS solutions were separated into three size fractions”.

#2-5 4. About the S275-295 analysis. In line 179, you said this value correlated with DOM molecular weight. Then, you said the value of HHS and MHS was lower than LHS. From my point, it is a matter of course since you separate them by filters. So my question is, what is the significance of this measurement? You said "the S275-295 values were similar to water-extracted organic matter, which ranged from 0.0074 to 0.029" in line 193, but that range is too wide for the discussion of similarity in my opinion.

Response #2-5: Thank you for the comments and suggestions.

The UV/Vis slope indicator S275-295 is a surrogate of the DOM molecular weight. The UV/Vis spectrum depends on the composition of HS/DOM. Generally, the different molecular weights have different S275-295 values. In this study, the HHS (10 kDa-0.45 μm) and MHS (1-10 kDa) had similar S275-295 values (0.0120±0.0004 and 0.0125±0.0014, respectively) and were significantly lower than the values of LHS (0.0165±0.0014). The SUVA254 and FI indicator values also had a similar tendency. Moreover, the log KHS of HHS and MHS had similar values, which were significantly higher than the LHS values. These results indicated that the HHS and MHS had similar compositions. Although the HHS and MHS had different molecular weights. they had similar S275-295 values, which suggested that the composition of HHS and MHS were similar. The significant correlation between HS and S275-295 suggested S275-295 was a good indicator to obtain the composition characteristic.

Line 193. We agree that the reported S275-295 values of water-extracted organic matter, which ranged from 0.0074 to 0.029, were too wide for the discussion of similarity. We have changed the comparison S275-295 to literature data as follows.

Lines 194-198. “The S275-295 values were higher than the alkaline-extracted organic matter, which ranged from 0.0074 to 0.0095, as reported by Hur et al. [5]. The S275-295 values were lower than the water-extracted organic matter, which ranged from 0.017 to 0.018, as reported by Xu et al. [32]. In addition, the S275-295 values of LMW DOM ranged from 0.019 to 0.028 and were significantly higher than the HMW DOM, which ranged from 0.009 to 0.012, as reported by Xu et al. [32].”

Reviewer 3 Report

Manuscript ijerph-653253 reports a well-done study and is of great interest. Manuscript has a correct structure and includes clear descriptions and explanations of the experimental design. The manuscript has valuable data. It presents data in an easy to follow way. The analytical approach is appropriate and the conclusions are coherent with the data and give some perspective for future studies. The style and the English language are fine, text requires a minor spell check.

Please, increase the quality and resolution of the figures.

Author Response

Modified Figs please see the attachment

Review #3

Comments and Suggestions for Authors

#3-1 Manuscript ijerph-653253 reports a well-done study and is of great interest. Manuscript has a correct structure and includes clear descriptions and explanations of the experimental design. The manuscript has valuable data. It presents data in an easy to follow way. The analytical approach is appropriate and the conclusions are coherent with the data and give some perspective for future studies. The style and the English language are fine, text requires a minor spell check.

Response #3-1: Thank you for the comments and suggestions. The spelling error has been corrected.

#3-2 Please, increase the quality and resolution of the figures.

Response #3-2: Thank you for the comments and suggestions.

We have modified Figs 2a-2c. We added more information as suggested by Reviewer #1-24. The modified Figs. 2a-2c are as follows.

Figure 2. Correlation of Phe log KHS with three optical indicators, (a) SUVA254 (b) S275-295 (c) FI.

Round 2

Reviewer 2 Report

The manuscript was improved by the attentive revision. I have no suggestion to this study.

However, double question marks remain in line 170. Is it appropriately?

I recommend to check this point.